# Impact of Inbreeding and Ancestral Inbreeding on Longevity Traits in German Brown Cows

**DOI:** 10.3390/ani13172765

**Published:** 2023-08-30

**Authors:** Anna Wirth, Jürgen Duda, Ottmar Distl

**Affiliations:** 1Institute of Animal Breeding and Genetics, University of Veterinary Medicine Hannover (Foundation), 30559 Hannover, Germany; anna.wirth@tiho-hannover.de; 2Landeskuratorium der Erzeugerringe für Tierische Veredelung in Bayern e.V. (LKV), 80687 München, Germany; juergen.duda@lkv.bayern.de

**Keywords:** inbreeding depression, ancestral inbreeding, longevity, German Brown cows

## Abstract

**Simple Summary:**

Observed increasing levels of inbreeding in German brown cattle imply the need to evaluate the resulting effects. If increasing levels of inbreeding come along with a reduction in the phenotypic performance, this is referred to as inbreeding depression. However, not all inbreeding is assumed to be unfavorable. In the context of inbreeding in combination with selection, it is presumed that negative variants can be eliminated from the population over time, implying ancestral inbreeding to be less harmful than new inbreeding. This is referred to as purging. To evaluate the effect of purging, ancestral inbreeding coefficients have been developed to allow to distinguish between new and ancestral inbreeding. The aim of this study was to evaluate inbreeding depression and purging for lifetime, lifetime performance, survival, and culling rates due to different reasons in German brown cows, by calculating the effects of classical and ancestral inbreeding coefficients. All longevity traits under study were affected by inbreeding depression. As the effects of ancestral inbreeding coefficients were significantly negative for lifetime and lifetime performance, while new inbreeding had no significant effect, there was no evidence of purging in the population under study. Thus, considering inbreeding levels in future mating plans helps to avoid a further decline in longevity due to inbreeding.

**Abstract:**

A recent study on the population structure of the German Brown population found increasing levels of classical and ancestral inbreeding coefficients. Thus, the aim of this study was to determine the effects of inbreeding depression and purging on longevity traits using classical and ancestral inbreeding coefficients according to Kalinowski (2002) (F_a_Kal_, F_New_), Ballou (1997) (F_a_Bal_), and Baumung (2015) (Ahc). For this purpose, uncensored data of 480,440 cows born between 1990 and 2001 were available. We analyzed 17 longevity traits, including herd life, length of productive life, number of calvings, lifetime and effective lifetime production for milk, fat, and protein yield, the survival to the 2nd, 4th, 6th, 8th, and 10th lactation number, and the culling frequencies due to infertility, or udder and foot and leg problems. Inbreeding depression was significant and negative for all traits but for culling due to udder and to foot and leg problems. When expressed in percentages of genetic standard deviations, inbreeding depression per 1% increase in inbreeding was −3.61 to −10.98%, −2.42 to −2.99%, −2.21 to −4.58%, and 5.13% for lifetime production traits, lifetime traits, survival rates, and culling due to infertility, respectively. Heterosis and recombination effects due to US Brown Swiss genes were positive and counteracted inbreeding depression. The effects of F_New_ were not significantly different from zero, while F_a_Kal_ had negative effects on lifetime and lifetime production traits. Similarly, the interaction of F with F_a_Bal_ was significantly negative. Thus, purging effects could not be shown for longevity traits in German Brown. A possible explanation may be seen in the breed history of the German Brown, that through the introgression of US Brown Swiss bulls ancestral inbreeding increased and longevity decreased. Our results show, that reducing a further increase in inbreeding in mating plans is advisable to prevent a further decline in longevity due to inbreeding depression, as purging effects were very unlikely in this population.

## 1. Introduction

Targeted selection programs in dairy cattle populations led to an enormous genetic progress, especially in milk production traits, over recent decades, at the cost of increasing levels of inbreeding as a result of focusing on a low number of highly selected elite bulls. German Brown is characterized by high milk production with high quality and superiority in length of productive life compared to Holsteins and German Fleckvieh. Introgression of US Brown Swiss bulls since 1966 contributed to the improvement in milk production traits in German Brown and, on the other hand, has led to increasing levels of individual inbreeding in animals with higher proportions of US Brown Swiss genes [1]. A previous study on milk performance for the first three lactation numbers of German Brown from the birth years between 1980 and 1992 indicated an inbreeding depression by 10 to 13 kg milk yield, 0.36 to 0.51 kg fat yield, and 0.35 to 0.47 kg protein yield per 1% increase in the inbreeding coefficient [2]. A slight tendency to lower inbreeding coefficients was seen in cows surviving the first three lactations compared to cows that left the herd before completing three lactations [2]. In Swiss Brown cows, first lactation and milk and fat yield with first calving between 1971 and 1986 were reduced by 26 and 0.08 kg per 1% of the inbreeding coefficient [3]. In Austrian Brown cows with first parturition between 1979 and 1991, estimates for inbreeding depression by 1% increase in the inbreeding coefficient were 6.3 to 10.4 kg milk yield, but close to zero for fat and protein percentages [4]. Length of productive life and lifetime energy-corrected milk yield decreased by 4.3 days and 165 kg per 1% inbreeding, respectively [4]. In addition, juvenile mortality of calves and heifers of Austrian Brown, born in the years 2001 to 2007, have been shown to be negatively affected by 0.49% per 1% increase in inbreeding [5]. Next to the effect of inbreeding depression, there is an ongoing debate about the existence of positive effects of inbreeding caused by inbred ancestors and referred to as purging [6,7]. Purging assumes that intensive selection may lead to an elimination of deleterious alleles from a population and therefore counteracting inbreeding depression [8]. This means that after a few generations of inbreeding, the best-performing individuals survive and reach the performance level of the non-inbred or less inbred individuals, or even achieve higher performance, while the poorer-performing individuals carrying deleterious alleles die or do not reproduce, so that these alleles are eliminated from the population [9,10]. According to Dickerson [11], the response to selection is stronger in mating systems with complete sibling mating than in systems with random mating. To date, the presence of purging has been studied in various experiments, including self-fertilization in plants [12], sibling mating in animals [13,14,15,16], by mainly examining traits related to life history [17], or in small captive populations [18,19]. For example, in a previous breeding experiment with quails, it was shown that populations with inbreeding in previous generations were less prone to inbreeding depression than populations without inbreeding history. Furthermore, the reproductive performance of populations with intensive inbreeding was better after a few generations than that of populations with random mating [13,20]. However, the intensity of purging depends on the extent to which a trait is affected by deleterious alleles and how these alleles affect fertility and survival. Lethal or semi-lethal alleles are more easily eliminated than less deleterious ones [8,9,17]. To analyze the effect of purging based on pedigree data, ancestral inbreeding coefficients have to be calculated [18,21,22]. Only few studies reported ancestral coefficients for different Holstein dairy cattle populations. The effects of the ancestral inbreeding coefficients on production, fertility, birth traits, and survival traits were not consistent [6,7,23,24,25]. For example, in Irish and Dutch Holstein evidence of the presence of purging effects for production traits could be shown [6,23], whereas results for Canadian and Iranian Holstein were not significant or even contradictory [24,25]. As the extent of inbreeding depression as well as purging may depend on different factors such as the population structure, the selection history, and the considered trait [8,10], it is crucial to analyze and monitor these effects in every population to adopt appropriate measures. For the German Brown population, classical and ancestral inbreeding coefficients have recently been reported and increasing trends with birth years were shown for classical and ancestral inbreeding coefficients [26]. The breed proportion of US Brown Swiss is increasing, but length of life and productive life were not positively correlated with increasing breed proportions of US Brown Swiss; therefore, this follow-up study in German Brown has been initiated with the objective to analyze inbreeding depression and purging effects on longevity traits including herd life (HL), length of productive life (LPL), lifetime milk production, and survival and culling incidences. We used an uncensored data set of Brown German cows born in the years 1990 to 2001. In order to account for the US Brown Swiss breed percentage, heterosis and recombination effects were considered simultaneously in animal models. 

## 2. Materials and Methods

The data were obtained from the official milk recording organization of Bavaria (Landeskuratorium der Erzeugerringe für tierische Veredelung in Bayern e.V., LKV). This data set comprised all German Brown with their available pedigrees and lifetime milk performance since 1990 [26]. For the analysis of longevity traits, all first-calving heifers born between 1990 and 2001 were regarded. All these animals had known parents as well as registered dates and reasons why they left the herd and, therefore, they had complete lifetime records. The final data set contained 480,440 cows with known parents and a corresponding pedigree file of 820,558 individuals. The data set was used to analyze measures of longevity, defined as functional, as they were corrected for the relative fat and protein yield of the single cow within their herd [27,28]. As suggested by Schuster et al. (2020) [29], herd life (HL) was defined as the time period from birth until culling. Length of productive life (LPL) included the time period from first calving to culling. Number of lactations (NC) was the sum of lactation numbers per lifetime which the respective cow reached. Lifetime milk yield (LMY), fat yield (LFY), and protein yield (LPY) were defined as total yield across all lactations. Lifetime efficiency for milk yield, (EffLMY), fat yield (EffLFY), and protein yield (EffLPY) were calculated as the ratio of lifetime production to herd life in days. Survival to lactation two (Surv1), four (Surv3), six (Surv5), eight (Surv7), and ten (Surv9) was calculated as the proportion of cows that reached the respective lactation number, e.g., cows that survived the first lactation and had at least a second calving date were encoded with Surv1 = 1 and all other cows with 0. Results are displayed as relative frequencies. The frequencies of the three most common culling reasons, namely culling due to foot and leg problems (Cul_CL_), infertility (Cul_INF_), and udder problems (Cul_UD_), were analyzed as binary traits, indicating that the cow did leave the herd due to this respective specific reason. Classical and ancestral inbreeding coefficients were available from our previous study and merged with the present data set including 448, 440 cows [26].

To account for the effect of crossbreeding with US Brown Swiss bulls starting in 1966, the coefficients of heterosis (HET = P_S_ (1 − P_D_) + P_D_ (1 − P_S_)) and recombination loss (REC = P_D_ (1 − P_D_) + P_S_ (1 − P_S_)) were defined based on the maternal (P_D_) and paternal (P_S_) breed proportion of Brown Swiss genes [30].

### Statistical Analysis

The effect of the different inbreeding coefficients was calculated by regressing phenotypic values on the inbreeding coefficient and accounting for animal and environmental effects. The following linear univariate animal model, parameterized for non-genetic effects according to Punsmann, Duda and Distl [28], was employed
Y_ijklmnopqr_ = µ + BY_i_ + BM_j_ + b_1_AFC_k_ + WHPL_l_ + HPL_m_ + b_2_HET_n_ + b_3_REC_o_ + b_4_INB_p_ + herd_q_ + animal_r_ + e_ijklmnopqr_,(1)
where Y_ijklmnopqr_ = dependent variate with HL, LPL, NC, LMY, LFY, LPY, EffLMY, EffLFY, EffLPY, Surv1, Surv3, Surv5, Surv7, Surv9, Cul_CL_, Cul_INF,_ or Cul_UD_. BY_i_ is the ith class of birth year for i = 1 (1990–1993), 2 (1994–1997), and 3 (1998–2001), BM_j_ is the jth class of birth month for j = 1–12, AFC_k_ = age at first calving in months and b_1_ the corresponding linear regression coefficient on the age at first calving, WHPL_l_ = within herd relative production level calculated as deviation of the sum of fat and protein production of each cow from the herd mean for fat and protein yield and expressed in standard deviation classes for l = 1–12, HPL_m_ = herd production level of fat and protein yield expressed in standard deviations for m-classes from 1 to 8, HET_n_ = coefficient of heterosis effects, REC_o_ = coefficient of recombination effects, INB_p_ = inbreeding coefficient F, F_a_Bal,_ or Ahc, b_2_, b_3_, and b_4_ are the linear regression coefficients of heterosis, recombination, and inbreeding effects, respectively, animal = random animal effect, herd = random herd effect, e = random error term.

The expected values for F, HET, and REC in the current population were calculated based on the observed estimates for the regression coefficients obtained from model 1 and the average coefficients of F, HET, and REC that were 0.018, 0.441, and 0.369, respectively. As F_a_Bal_ does not consider the probability that an individual is identical by descent (IBD) itself, we included the interaction of F_a_Bal_ with F as suggested by Ballou (1997) [18] in model 2
Y_ijklmnopqrs_ = µ + BY_i_ + BM_j_ + b_1_AFC_k_ + WHPL_l_ + HPL_m_ + b_2_HET_n_ + b_3_REC_o_ + b_4_F_p_ + b_5_F×F_a_Balq_ + herd_r_ + animal_s_ + e_ijklmnopqrs_(2)
where fixed and random effects were defined as above and F is the coefficient of inbreeding, F×F_a_Bal_ = interaction of the coefficient of inbreeding with the ancestral inbreeding coefficient according to Ballou [18], and b_4_ and b_5_ are the corresponding linear regression coefficients.

Model 3 included the effects of either Ahc (ancestral history coefficient according to Baumung, Farkas, Boichard, Mészáros, Sölkner and Curik [22]) or F_a_Bal_ (ancestral inbreeding coefficient according to Ballou [18]) as F_ANC_ together with F simultaneously:Y_ijklmnopqrs_ = µ + BY_i_ + BM_j_ + b_1_AFC_k_ + WHPL_l_ + HPL_m_ + b_2_HET_n_ + b_3_REC_o_ + b_4_F_p_ + b_6_ F_ANCq_ + herd_r_ + animal_s_ + e_ijklmnopqrs_(3)

In model 4 the effect of the inbreeding coefficients according to Kalinowski, Hedrick and Miller [21] were analyzed by considering F_a_Kal_ and F_New_ simultaneously in the model:Y_ijklmnopqrs_ = µ + BY_i_ + BM_j_ + b_1_AFC_k_ + WHPL_l_ + HPL_m_ + b_2_HET_n_ + b_3_REC_o_ + b_4_F_a_Kalp_ + b_5_F_Newq_ + herd_r_ + animal_s_ + e_ijklmnopqrs_(4)

Variance components were estimated using VCE 6.0.2 [31], and were used for estimation of random and fixed effects as well as linear regressions using PEST, version 4.2.6. Further statistical analyses were performed in SAS, version 9.4 (Statistical Analysis System, Cary, NC, USA, 2023). The model employed for the estimation of heritabilities included all fixed and random effects models as the above-mentioned models, and the proportion of US Brown Swiss in classes of 10% (<31%, 31–40%, 41–50%, 51–60%, 61–70%, 71–80%, >80%) [21]. Animal models with a binomial distribution function for 0/1-traits yielded very similar estimates for heritabilities, and thus, we employed linear animal models.

## 3. Results

Phenotypic means of the longevity measures of German Brown cows born between 1990 and 2001 are presented in Table 1. German Brown cows reached an average age of 6.15 years and were productive for 56.2% (3.46 years) of their lifetime. The 95% quantiles for HL and LPL were 11.16–20.98 and 8.48–18.09 years with 8–17 calvings, respectively. Quartiles for LMY were from 7537 to 32,736 kg milk and the top 5% cows had an LMY of 58,065–191,049 kg milk. For LFY and LPY, the top 5% cows produced 2418–8045 kg fat and 2075–6787 kg protein, respectively.

Heritability estimates for HL, LPL, and NC were around 0.16 (Table 2). Effective lifetime production showed higher heritabilities (0.128–0.154) compared to lifetime production (0.126–0.135). For survival traits, heritability was highest for Surv5 (0.088) and lowest for Surv1 (0.025), whereas heritability for the three most common culling reasons ranged from 0.034 to 0.055. Heritability estimates with their corresponding standard errors for Surv1, Surv3, Surv5, Surv7, and Surv9 were 0.048 (0.005), 0.111 (0.006), 0.182 (0.007), 0.282 (0.011), and 0.319 (0.024), and for Cul_CL_, Cul_INF_, and Cul_UD_ 0.147 (0.008), 0.063 (0.005), and 0.099 (0.007), respectively, when transformed to the underlying scale with the formula of Dempster and Lerner [32].

Pedigree completeness was measured through the number of complete equivalent generations (GE). The GE was on average 5.78 with an increasing trend from 4.54 in 1990 to 6.13 in 2001. The average individual inbreeding coefficient for the reference population was 0.018 and increased over time from 0.013 in 1990 to 0.024 in 2001. Of the 480,440 cows, 88.5% were inbred, with the highest coefficient of individual inbreeding being 0.3175. The mean ancestral coefficients of inbreeding ranged from 0.001 to 0.012 and showed an increasing trend over time, with the highest increase observed for F_a_Bal_ and Ahc, which increased from about 0.013 in 1990 to 0.019 in 2001 (Appendix A). Correlation between F_a_Bal_ and Ahc was 0.99. The lowest correlations (0.07) were found between F_New_ and F_a_Bal_ as well as Ahc. Correlations between coefficients of heterosis and coefficients of inbreeding (F), F_New,_ F_a_Kal_, F_a_Bal_, and Ahc were −0.33, −0.07, −0.06, −0.11 and −0.11, respectively (Appendix A).

The results from model 1 revealed a significant unfavorable effect of F on all longevity traits studied, except for Cul_CL_ and Cul_UD_ (Table 3). All estimates from linear regression coefficients refer to 100% inbreeding and consider the difference between a non-inbred and a fully inbred animal. A 1% increase in inbreeding reduces HL and LPL by 7.3 and 7.7 days, respectively. Regarding survival, inbreeding depression was higher for cows with less lactation numbers than for cows with more lactation numbers and was highest for Surv3 (Table 3). When expressed in percentages of genetic standard deviations, lifetime production traits were more influenced by inbreeding than HL, LPL, NC, and survival to the following lactation number (Surv1 to Surv9), which had decreasing inbreeding depression when cows reached higher lactation numbers (Appendix A). Inbreeding significantly increased the frequency of Cul_INF_, with an increase by 0.399 per 100% inbreeding. Significant positive heterosis effects were observed for most traits with 100% heterosis increasing HL by 161 days and LMY by 1700 kg, and increasing Surv3 and Surv5 by 6.8% and 5.9%, respectively. Recombination effects were also significantly positive for HL, LPL, NC, and Surv1 to Surv7. No significant heterosis and recombination effects were found for effective lifetime production traits (Table 3). Regarding the expected cumulated effects of inbreeding, heterosis, and recombination for the current population, the combined effects were favorable for all traits apart from effective lifetime production and culling due to infertility. Projecting these effects to the birth year 2014, based on coefficients from 2014, heterosis and recombination effects decrease, while inbreeding effects increase, leading to expected decreasing combined effects and even to more unfavorable negative effects for LPY and effective lifetime production (Appendix A).

Results for the effects of ancestral inbreeding coefficients are presented in Table 4 and Table 5. Using model 2, the regression coefficients of the interaction of F×F_a_Bal_ were significantly negative for HL, LPL, NC, lifetime, and effective lifetime production traits and the corresponding effects of F were slightly reduced compared with the results of model 1.

Significant negative effects of F_a_Kal_ were found for HL, LPL, NC, Surv3, and lifetime production traits using model 4, whereas the effect of F_New_ was close to zero and not significantly different from zero for these traits (Table 4).

Considering Ahc, negative regression coefficients were observed for HL, LPL, NC, lifetime production, and Surv3 and 5. Estimates for regression coefficients from model 4 were similar to the estimates obtained from model 1 (Table 5). Moreover, the regression coefficients for Ahc equaled those of F_a_Bal_ in the corresponding models and, therefore, only the results for Ahc are presented.

The expected phenotypes of highly (95% percentile) and lowly (5% percentile) inbred cows were calculated to further illustrate the effect of changing the inbreeding coefficients. For classical inbreeding, the differences corresponded to 33 days for HL and 873 kg for LMY. Differences in ancestral inbreeding levels were lower with 7 and 11 days of HL and 125 and 174 kg of LMY for F_a_Kal_ and Ahc, respectively (Appendix A).

## 4. Discussion

Inbreeding depression was obvious for all traits analyzed but for Cul_CL_ and Cul_UD_. On the other hand, all longevity traits and lifetime performance traits but lifetime efficiency showed significantly positive heterosis effects through crossbreeding with US Brown Swiss bulls. Recombination effects were also positive but smaller than heterosis effects and had importance for longevity traits only. Models regarding ancestral inbreeding coefficients revealed significant negative effects, and thus we were not able to demonstrate purging effects. Under purging, ancestral inbreeding would have been expected to exert positive effects on the traits.

The negative effect of inbreeding on LPL of 7.7 (7.70 = 2.11 years × 365 days/100) days per 1% inbreeding was slightly higher than estimates reported for Austrian Brown, Simmentals including Simmental x Red Friesian crosses, and pure Simmentals with −4.3, −5.7, and −4.9 days per 1% inbreeding, respectively [4]. In a US American study for 2.6 million Holsteins from 1983 to 1993, LPL was reduced by 5.96 days per 1% inbreeding [33].

Lifetime production traits were less negatively affected by inbreeding in Austrian Simmental, Simmental x Red Friesian, and Brown Swiss compared to this study. In the latter breeds, fat-corrected milk was reduced by −136.7, −109.9, and −165.0 kg per 1% inbreeding, respectively [4]. Similarly, US Holsteins showed a decrease due to 1% inbreeding of −177.17 kg, −6.01 kg, and −5.45 kg lifetime milk, and fat and protein yield, respectively [33]. Nevertheless, lifetime was limited in the Austrian study to 10 lactations [4] and in the US study to 84 months after first calving [33], whereas our study included all records until the cows left the herd.

The higher impact of inbreeding on lifetime performance compared to longevity traits, HL, LPL, and NC, becomes obvious when expressed in percentages of their phenotypic and genetic standard deviations. This may indicate that the reduced lifetime production in German Brown cattle in this study is not only due to the shortened LPL, but the production traits themselves may be negatively influenced by inbreeding. This is in agreement with previous studies on different dairy cattle populations that reported inbreeding depression for milk performance in the first three lactations [6,23,24,25]. The highest effects due to inbreeding depression, expressed in percentages of the genetic standard deviations, were demonstrated for LMY, Surv1, and EffLFY. Therefore, inbreeding depression may have the largest negative effects on the expected breeding progress of these traits.

In order to evaluate contributions from US Brown Swiss genes, we also employed a model with F and its interaction with classes of US Brown Swiss genes. These analyses showed, for all traits under analysis, that cows with less than 50% US Brown Swiss genes had less inbreeding depression and inbreeding depression increased up to 70–80% US Brown Swiss genes and then decreased with even higher proportions of US Brown Swiss genes. This may indicate that negative effects of inbreeding seem related with proportions of US Brown Swiss genes and the strategy of using bulls with different proportions of US Brown Swiss genes. Introducing US Brown Swiss bulls may be associated with less inbreeding depression, because these US Brown Swiss bulls may not be so closely related with the German Brown population in comparison to German Brown bulls with less than 100% US Brown Swiss genes.

Inbreeding depression may have caused a higher risk for younger cows to be culled. Cows which survived more lactations may be assumed to be prone to a lower risk to be culled due to the effects on inbreeding depression. Thus, cows surviving to higher lactation numbers are expected to show decreasing inbreeding coefficients (Appendix A). The decreasing inbreeding coefficients in cows reaching a higher lactation number reduce the size of inbreeding depression in these cows. Selection of cows for following lactation numbers may also lower the increase in inbreeding and the amount of inbreeding depression in the next generation. Survivors to higher lactations may have more progeny than cows leaving the herd early and therefore, increase in inbreeding and inbreeding depression may be smaller compared to a situation when all cows have the same chance to produce progeny. The regression coefficients were highest for Surv3, implying that an inbred cow is at highest risk to leave the herd before the fourth lactation. As this is also the period of time when cows leave the herd on average, i.e., with 3.51 calvings and an LPL of 3.46 years in the studied population, this further supports the lifetime-limiting effect of inbreeding. Nevertheless, the effect of inbreeding on survival was small, as it would explain a reduction of 0.8% (−0.8% = (0.018) × (−0.457) ×100) for Surv3, in relation to the average degree of inbreeding in the population studied. With respect to survival to the second lactation, a comparable regression coefficient of −0.3 was reported for Irish Holstein based on 42,723 survival records [23]. The mean level of inbreeding was higher in Irish Holstein with F = 0.0268 than in German Brown with 0.018 though. A decrease in survival from the second through fifth and sixth lactation in comparison to lactation one was observed in American Holstein and Jersey, respectively, with increasing inbreeding [34,35]. When considering the stayability to 48 months of age, the effects of inbreeding in US Ayrshire, Guernsey, Holstein, Jersey, and Brown Swiss ranged from −0.011 to −0.002 in a linear sire model and were closer to zero than the estimates for Surv1 and Surv3 in this study [36]. Because of the high standard errors of 0.0017, the authors concluded that inbreeding had no negative effect on stayability to 48 months [36]. However, it has to be considered that the degree of inbreeding in this study was as low as 0.009. When longevity was expressed as relative culling risk, small but significant effects were found in Canadian Holstein, Jersey, and Ayrshire compared with non-inbred cows. The effect was greater for cows in inbreeding classes beyond 12.5% [37]. In US Jersey cows that calved for the first time between 1981 and 2000, a slightly higher culling risk was observed in animals with an inbreeding coefficient greater than 10% than in animals with an inbreeding coefficient less than 5% [38]. Regarding the reasons why cows left the herd, the increase in the frequency of culling due to infertility was related with inbreeding depression. Infertility is the most common reason in German Brown cows for leaving the herd and is one of the most common reasons in dairy cattle populations [39,40,41]. The proportion of cows leaving the herd due to infertility has already been shown to be highest in lower lactation numbers in German Brown [42] and Australian dairy cattle [41]. Cows surviving fewer lactations were more inbred in this study than cows surviving higher lactations; thus, it can be concluded that German Brown cows with higher levels of inbreeding are more likely to be affected by infertility, which is in agreement with the results on Spanish Holsteins, where cows with high inbreeding coefficients (6.25–12.5%) have a 1.68% lower pregnancy rate than non-inbred cows [43]. In general, negative effects of inbreeding on fertility have already been reported for different dairy cattle populations [6,23,43,44,45]. Previous studies reported positive correlations between fertility and longevity traits, further explaining the link between inbreeding, longevity, and fertility [46,47]. Thus, preventing inbreeding may improve fertility and reduce culling due to infertility, which would lead to an increasing HL and LPL [42].

We concurrently considered heterosis and recombination effects because the German Brown is a crossbred population of Original Brown and US Brown Swiss with an increasing trend of breed proportions of US Brown Swiss [26]. Considering the effects in relation to the mean heterosis and inbreeding coefficients of the population under study, heterosis effects are positive and are counteracting the negative impact of inbreeding. Using population averages, heterosis is expected to increase HL by 70 days, while inbreeding decreases HL by 13 days. Similar expectations were found for lifetime production traits. Heterosis estimates for Surv1-5 in this study range from 3.4% to 6.8% and are in the range of the results from studies of crossbred populations involving various dairy cattle breeds from New Zealand [48], Sweden [49], and Denmark [50]. The introduction of US Brown Swiss genes was performed with only a small number of sires that were mated frequently within the population. Thus, the high breed proportion of the today’s German Brown population stems from a small gene pool of US Brown Swiss genetics [26]. This means that crossing with US Brown Swiss sires led to increasing levels of inbreeding, as co-ancestries between cows rapidly increased in the first crossbred generations [26]. This may furthermore explain the positive and significant recombination effects for HL, LPL, NC, and survival. When summarizing the effects of heterosis, recombination, and inbreeding, the positive combined effects suggest a balanced contribution of inbreeding and the introduction of new genes of US Brown Swiss. When projecting these results to the 2014 birth cohort, based on the data of our previous study [26], the combined effects for all traits decrease, mainly due to the increasing trend in inbreeding, and the increasing breed proportion of US Brown Swiss is expected to lead to decreasing heterosis effects. Nevertheless, it has to be considered that estimates of inbreeding effects may change over time, e.g., due to purging; thus, the effects for future generations may deviate from the extrapolation of the current estimates.

The breeding history of the German Brown is also important for the interpretation of ancestral inbreeding effects and purging. So far, there are only a few studies having analyzed the effects of ancestral inbreeding coefficients in dairy cattle populations, whereby longevity traits have rarely been under study (Appendix A). In German Brown cattle, the effects of ancestral inbreeding coefficients were negative and significant for lifetime and lifetime performance traits. None of the models accounting for ancestral inbreeding coefficients revealed evidence of purging. Considering the definition of Ballou (1997) [18], a significant favorable regression coefficient of the interaction of F with F_a_Bal_ would provide evidence for purging, while this study revealed significant negative effects for HL, LPL, NC, and lifetime production. Moreover, the effects of F_a_Bal_ were also negative when considered in the model as a regression coefficient alone. In Irish Holsteins, a positive, but not significant interaction of F with F_a_Bal_ was observed for survival from first to second lactation and 305-day production traits, while the effect of the simple regression model with F_a_Bal_ was significantly positive only for milk and protein yield [23].

The approach with Ahc assumes that alleles that have met more frequently in the past are less likely to have a detrimental effect than alleles that have never or only a few times before been identical-by-descent (IBD) [22]. Since the inbreeding coefficient indicates how often a randomly selected allele was IBD during pedigree segregation, an increasing Ahc could be associated with a beneficial effect on phenotype [6,22]. In Dutch Holsteins, positive effects for Ahc were reported for first lactation protein yield [6], whereas the current study showed only negative effects for Ahc on lifetime and lifetime production traits. In addition, inclusion of Ahc alone or simultaneously with F resulted in similar estimates for the regression coefficients of Ahc. We also tested the interaction of Ahc and F using model 2, which revealed comparable results to the interaction of F_a_Bal_ and F. In general, results for F_a_Bal_ and Ahc were very similar in this study with slightly higher estimates of F_a_Bal_ for some of the traits when included in the different models, which may indicate equal meaning of these ancestral inbreeding measures, at least for the current pedigree structure of German Brown cows, where correlation of F_a_Bal_ and Ahc was close to one [26,51].

According to Kalinowski et al. (2000) [21], successful purging would be present if F_New_ had a more detrimental effect on the trait compared to F_a_Kal_. In contrast, in this study the effects of F_New_ were close to zero and not significant, while F_a_Kal_ had significantly negative effects on HL, LPL, NC, and lifetime production. These results indicate that the detrimental effects of inbreeding for longevity traits are mainly due to ancestral generations. In Irish Holsteins, a significantly negative effect of F_a_Kal_ was reported for survival to second lactation, while F_New_ was not significant [23]. For 305-day production traits, F_New_ was responsible for greater losses than F_a_Kal_ in Irish Holsteins [23]. Also, in Dutch Holsteins, F_New_ was significantly negatively associated with first lactation yield [6]. No significant effects of F_a_Kal_ and F_New_ were observed in Canadian Holsteins for lactation yield, but for milk and protein yield the effects of F_New_ were unfavorable while those from F_a_Kal_ were favorable [24].

Our results for German Brown using different models to account for ancestral inbreeding were consistent and did not show favorable effects of inbreeding from more ancient generations for lifetime and lifetime performance traits. Therefore, we assume that purging effects are very unlikely. However, the informative value of the different models used strongly depends on the structure of the pedigree. Deeper pedigrees are likely to reveal more inbreeding, which could also affect the results in terms of inbreeding depression and purging. The depth of the pedigree in this study was comparable to other studies, as we have discussed previously [19].

There may be a few reasons why purging was not detected in the German Brown population for longevity traits, but instead negative genetic contributions of the ancestral generations were discovered. First, the efficiency of purging depends on the structure of inbreeding within the population and the rate at which inbreeding increases [8] implying differences in the success due to different breeding and selecting histories. The overall level of inbreeding and ancestral inbreeding in the current population was not high due to introgression of US Brown Swiss bulls; thus, purging might possibly occur in future generations, where a further increase in inbreeding has been observed [26]. Additional analysis of survival traits using birth year cohorts from 2002 to 2008 and records through 2018 for F_a_Kal_ and F_New_ gave estimates for Surv1 of 0.0998 ± 0.3412 and −0.1168 ± 0.0522 with *p*-Values of 0.7698 and 0.0251, respectively, and for Surv3 estimates of 0.0159 ± 0.4229 and −0.11515 ± 0.0646 with *p*-Values of 0.9699 and 0.07488, respectively. Mean values and standard deviations for F_a_Kal_ and F_New_ were 0.00183 ± 0.00264 and 0.01569 ± 0.01695, respectively. These preliminary data suggest that significant purging effects are not yet present, even though positive effects for F_a_Kal_ were observed. On the other hand, estimates for F (Surv1: −0.2790 ± 0.0438, *p*-Value < 0.001 and Surv3: −0.5821 ± 0.0548, *p*-Value < 0.001) and F_New_ indicated larger negative effects due to inbreeding depression. A similar picture emerged for effective lifetime performance for milk, fat, and protein yield in this data set with positive non-significant estimates for F_a_Kal_ and negative non-significant estimates for F_New_. Purging may not even counterbalance the negative effects of new inbreeding in these data from 2002 to 2008. Other opportunities for including survival in more recent data would be multiple trait animal models that only regard survival up to the first four lactation numbers, but split the first or each lactation into three different periods to account for the distribution of reasons for culling in the different lactation periods. These multiple trait animal models need to be tested to see whether they can be extended to cows surviving more than seven or nine lactations if records from the younger birth cohorts are incomplete [52,53,54].

Increase in inbreeding in the ancestral generations was mainly due to multiple use of a limited number of bulls. In the German Brown population introgression with US Brown Swiss bulls started in 1966 resulting in higher milk production at the cost of a steeper increase in inbreeding [26] and reduced lifetime [28]. Thus, selecting towards higher milk production favored the more intense use of US Brown Swiss genetics. Sufficient purging is characterized through an elimination of deleterious genes. Through the introduction of US Brown Swiss, more genes were introduced that negatively influenced longevity and inbreeding depression did not decline through purging and elimination of unfavorable genes associated with longer lifetime in more inbred animals. Thus, it is likely that the negative ancestral effects on longevity traits might predominantly stem from ancestors representing a higher breed proportion of US Brown Swiss.

Furthermore, the greatest success of purging has been found under high selection pressure [9]. In the German Brown, the productivity traits fat and protein yield and protein percentage were economically weighted with 48% until 2015, resulting in a high selection pressure for these traits. In contrast, economical weights for functional traits such as longevity and fertility were only 16.1% and 8.6%, indicating less intensive selection. Estimation of breeding values for longevity started in 1995, so these traits had less time to express purging. In addition, purging depends on the genetic structure of the trait [7]. Lower heritabilities for functional traits compared to production traits make selection less successful than for productivity traits. To date, ancestral inbreeding effects on fertility were mostly not significant or at least less conclusive considering purging [6,23,24]. Thus, assuming that both fertility and longevity are functional traits subject to lower selection pressure, our results are consistent with these previous findings.

## 5. Conclusions

Significant inbreeding depression was shown in German Brown cows for lifetime, lifetime production, and effective lifetime production traits. However, compared to the current level of inbreeding, the negative effects through inbreeding depression are counterbalanced by positive heterosis and recombination effects, leading to positive combined effects. We were not able to demonstrate purging as all models accounting for ancestral inbreeding did not reveal positive effects on lifetime, lifetime production, survival, and main culling reasons for F_a_Kal,_ F_a_Bal,_ Ahc, and F×F_a_Bal_. Inbreeding depression was predominantly caused by ancestors from more ancient generations than inbreeding in newer generations. The intensive use of US Brown Swiss bulls in the first generations after immigration started may also have contributed these outcomes. In addition, longevity traits have been under less selection pressure at this time, thus positive effects due to selection against inbreeding depression are not likely to appear in the data of the present study. Thus, it is worth considering the dynamics of the different ancestral inbreeding coefficients in future generations. As ancestral inbreeding is increasing more strongly than new inbreeding in German Brown cows to recent birth years, purging effects for longevity can possibly appear in future generations. Our results indicate that a further decline of lifetime and lifetime production can be prevented when measures can be implemented to slow down the increase in ancestral inbreeding and the further increase in new inbreeding through US Brown Swiss genes. At present, our data have some limitations in terms of the depth of the pedigree and the endpoint of the data (2001), future studies should possibly use censored data and survival analysis methods or multiple trait animal models for survival to different parities and periods within the respective lactations to gain insights into the actual development of longevity traits.

## Figures and Tables

**Table 1 animals-13-02765-t001:** Phenotypic means, standard deviations (SD), and 75% confidence intervals (75% CI) of herd life (HL), length of productive life (LPL), number of calvings (NC), lifetime milk yield (LMY), lifetime fat yield (LFY), lifetime protein yield (LPY), effective lifetime milk yield (EffLMY), effective lifetime fat yield (EffLFY), effective lifetime protein yield (EffLPY), survival to 2nd (Surv1), 4th (Surv3), 6th (Surv5), 8th (Surv7), and 10th (Surv9) lactation, and culling rate due to claw and leg disorders (Cul_CL_), infertility (Cul_INF_), and udder diseases (Cul_UD_) in 480,440 German Brown cows.

Trait	Mean	SD	75% CI
HL (years)	6.15	2.65	4.01–7.78
LPL (years)	3.46	2.63	1.31–5.08
NC	3.51	2.32	2–5
LMY (kg)	22,319	18,445	7537–32,736
LFY (kg)	933	769	316–1368
LPY (kg)	797	661	269–1169
EffLMY (kg/day)	8.420	4.359	5.092–11.555
EffLFY (kg/day)	0.352	0.184	0.213–0.484
EffLPY (kg/day)	0.301	0.159	0.181–0.414
Surv1	0.760	0.427	
Surv3	0.429	0.495	
Surv5	0.193	0.394	
Surv7	0.066	0.248	
Surv9	0.017	0.128	
Cul_CL_	0.118	0.322	
Cul_INF_	0.252	0.434	
Cul_UD_	0.115	0.319	

**Table 2 animals-13-02765-t002:** Estimates of phenotypic variance (σ^2^_p_), additive genetic variance (σ^2^_a_), herd variance(σ^2^_herd_), residual variance (σ^2^_e_), and heritabilities (h^2^) with their corresponding standard errors (SE) for herd life (HL), length of productive life (LPL), lifetime milk yield (LMY), lifetime fat yield (LFY), lifetime protein yield (LPY), effective lifetime milk yield (EffLMY), effective lifetime fat yield (EffLFY), effective lifetime protein yield (EffLPY), survival to 2nd (Surv1), 4th (Surv3), 6th (Surv5), 8th (Surv7), and 10th (Surv9) lactation, and culling rate due to foot and leg problems (Cul_CL_), infertility (Cul_INF_), and udder diseases (Cul_UD_).

Trait	σ^2^_p_	σ^2^_a_	±SE	σ^2^_herd_	±SE	σ^2^_e_	±SE	h^2^ ± SE
HL	4.798	0.770	±0.020	0.498	±0.008	3.529	±0.015	0.160 ± 0.004
LPL	4.780	0.758	±0.020	0.482	±0.008	3.541	±0.015	0.159 ± 0.004
NC	4.323	0.689	±0.018	0.368	±0.007	3.266	±0.014	0.159 ± 0.004
LMY (10^6^)	212.000	2.858	±0.861	17.050	±0.304	166.500	±0.640	0.135 ± 0.004
LFY (10^3^)	359.332	45.418	±1.423	26.620	±0.476	287.294	±1.102	0.126 ± 0.004
LPY (10^3^)	265.901	33.958	±1.054	20.709	±0.363	211.234	±0.810	0.128 ± 0.004
EffLMY (10^−2^)	764.458	117.675	±3.471	67.842	±1.176	578.940	±2.585	0.154 ± 0.004
EffLFY (10^−2^)	1.277	0.164	±0.057	0.101	±0.002	1.013	±0.004	0.128 ± 0.004
EffLPY (10^−2^)	0.964	0.132	±0.004	0.088	±0.001	0.744	±0.003	0.137 ± 0.004
Surv1 (10^−3^)	128.340	3.246	±0.312	5.989	±0.123	119	±0.327	0.025 ± 0.002
Surv3 (10^−3^)	202.640	14.114	±0.751	12.201	±0.227	176.320	±0.653	0.070 ± 0.004
Surv5 (10^−3^)	144.590	12.703	±0.501	8.087	±0.160	123.800	±3.363	0.088 ± 0.003
Surv7 (10^−3^)	60.628	4.566	±0.188	2.357	±0.0516	53.704	±0.167	0.075 ± 0.003
Surv9 (10^−3^)	16.467	0.555	±0.042	0.325	±0.010	15.588	±0.044	0.034 ± 0.003
Cul_CL_ (10^−3^)	104.560	5.790	±0.332	3.564	±0.083	95.209	±0.288	0.055 ± 0.003
Cul_INF_ (10^−3^)	179.200	6.051	±0.505	8.163	±0.171	164.980	±0.484	0.034 ± 0.003
Cul_UD_ (10^−3^)	102.200	3.739	±0.275	2.723	±0.073	95.742	±0.275	0.037 ± 0.003

Exponential terms in brackets relate only to variances.

**Table 3 animals-13-02765-t003:** Regression coefficient of inbreeding, heterosis, and recombination loss on herd life (HL), length of productive life (LPL), lifetime milk yield (LMY), lifetime fat yield (LFY), lifetime protein yield (LPY), effective lifetime milk yield (EffLMY), effective lifetime fat yield (EffLFY), effective lifetime protein yield (EffLPY), survival to 2nd (Surv1), 4th (Surv3), 6th (Surv5), 8th (Surv7), and 10th (Surv9) lactation number, and culling rate due to foot and leg problems (Cul_CL_), infertility (Cul_INF_), and udder diseases (Cul_UD_) estimated with model 1.

	F	SE	HET	SE	REC	SE
HL (years)	−1.99 ***	0.19	0.44 ***	0.05	0.16 *	0.07
LPL (years)	−2.11 ***	0.19	0.42 ***	0.05	0.16 *	0.07
NC	−2.48 ***	0.19	0.31 ***	0.05	0.13 *	0.07
LMY (kg)	−18,556 ***	1318	1700 ***	332	374	477
LFY (kg)	−791 ***	55	66.8 ***	13.7	19.9	19.6
LPY (kg)	−666 ***	46.8	50.5 ***	11.7	10.0	16.9
EffLMY (kg/day)	−3.916 ***	0.248	0.064	0.063	−0.128	0.091
EffLFY (kg/day)	−0.170 ***	0.010	0.000	0.003	−0.005	0.004
EffLPY (kg/day)	−0.140 ***	0.009	−0.001	0.002	−0.006 *	0.003
Surv1	−0.261 ***	0.033	0.034 ***	0.008	0.022 *	0.011
Surv3	−0.457 ***	0.042	0.068 ***	0.010	0.037 **	0.014
Surv5	−0.377 ***	0.035	0.059 ***	0.009	0.031 **	0.012
Surv7	−0.154 ***	0.023	0.025 ***	0.006	0.016 *	0.008
Surv9	−0.052 ***	0.012	0.009 ***	0.003	−0.002	0.004
Cul_CL_	−0.097 **	0.030	−0.010	0.007	−0.040 ***	0.010
Cul_INF_	0.399 ***	0.040	0.034 ***	0.009	0.036 **	0.013
Cul_UD_	−0.125 ***	0.030	−0.003	0.007	−0.010	0.010

*** *p* < 0.001, ** *p* < 0.01, * *p* < 0.05.

**Table 4 animals-13-02765-t004:** Regression coefficient of inbreeding (F), the interaction of the coefficient of inbreeding and ancestral inbreeding according to Ballou (F × F_a_Bal_) (model 2), ancestral inbreeding according to Kalinowski (F_a_Kal_), and new inbreeding according to Kalinowski (F_New_) (model 4) for herd life (HL), length of productive life (LPL), lifetime milk yield (LMY), lifetime fat yield (LFY), lifetime protein yield (LPY), effective lifetime milk yield (EffLMY), effective lifetime fat yield (EffLFY), effective lifetime protein yield (EffLPY), survival to 2nd (Surv1), 4th (Surv3), 6th (Surv5), 8th (Surv7), and 10th (Surv9) lactation, and culling rate due to foot and leg problems (Cul_CL_), infertility (Cul_INF_), and udder diseases (Cul_UD_) for German Brown cows.

	F	SE	F × F_a_Bal_	SE	F_a_Kal_	SE	F_New_	SE
HL (years)	−1.78 ***	0.21	−30.30 **	10.51	−7.06 *	2.90	−0.08	0.24
LPL (years)	−1.90 ***	0.21	−29.89 **	10.51	−6.90 *	2.90	−0.09	0.24
NC	−2.31 ***	0.20	−25.33 *	10.08	−6.61 *	2.78	−0.08	0.23
LMY (kg)	−16,943 ***	1408	−232,257 **	71,165	−40,527 *	19,640	118	1610
LFY (kg)	−723 ***	58	−9778 ***	2944	−1605 *	812	8.685	67.61
LPY (kg)	−607 ***	50	−8516 ***	2526	−1426 *	697	6.104	57.18
EffLMY (kg/day)	−3.668 ***	0.265	−35.615 **	13.394	−6.953	3.696	0.188	0.303
EffLFY (kg/day)	−0.160 ***	0.011	−1.472 **	0.553	−0.285	0.153	0.011	0.013
EffLPY (kg/day)	−0.131 ***	0.009	−1.309 **	0.477	−0.241	0.132	0.008	0.011
Surv1	−0.255 ***	0.036	−0.843	1.814	−0.184	0.501	−0.021	0.041
Surv3	−0.437 ***	0.044	−2.859	2.249	−1.491 *	0.621	−0.014	0.051
Surv5	−0.349 ***	0.038	−4.102 *	1.898	−0.953	0.524	−0.003	0.043
Surv7	−0.139 ***	0.025	−2.237	1.241	−0.240	0.343	−0.013	0.028
Surv9	−0.042 **	0.013	−1.369 *	0.656	−0.310	0.181	−0.015	0.015
Cul_CL_	−0.101 **	0.032	0.561	1.639	0.095	0.453	−0.055	0.037
Cul_INF_	0.377 ***	0.042	3.225	2.142	0.098	0.592	0.051	0.048
Cul_UD_	−0.119 ***	0.032	−0.814	1.629	0.624	0.450	−0.082 *	0.037

*** *p* < 0.001, ** *p* < 0.01, * *p* < 0.05.

**Table 5 animals-13-02765-t005:** Regression coefficients on the ancestral history coefficient (Ahc) according to Baumung for herd life (HL), length of productive life (LPL), lifetime milk yield (LMY), lifetime fat yield (LFY), lifetime protein yield (LPY), effective lifetime milk yield (EffLMY), effective lifetime fat yield (EffLFY), effective lifetime protein yield (EffLPY), survival to 2nd (Surv1), 4th (Surv3), 6th (Surv5), 8th (Surv7), and 10th (Surv9) lactation, and culling rate due to foot and leg problems (Cul_CL_), infertility (Cul_INF_), and udder diseases (Cul_UD_) for German Brown Cows.

	Only Ahc in Model ^1^		Ahc and F Simultaneously in Model ^2^			
	A_hc_	SE	F	SE	A_hc_	SE
HL (years)	−0.88 **	0.28	−1.99 ***	0.19	−0.88 **	0.28
LPL (years)	−0.87 **	0.28	−2.11 ***	0.19	−0.88 **	0.28
NC	−0.70 **	0.27	−2.48 ***	0.19	−0.70 *	0.27
LMY (kg)	−4866 *	1904	−18,559 ***	1318	−4888 *	1904
LFY (kg)	−169 *	79	−791 ***	55	−169 *	79
LPY (kg)	−160 *	68	−666 ***	47	−161 *	68
EffLMY (kg/day)	−0.660	0.358	−3.916 ***	0.248	−0.665	0.358
EffLFY (kg/day)	−0.018	0.015	−0.170 ***	0.010	−0.018	0.015
EffLPY (kg/day)	−0.019	0.013	−0.140 ***	0.009	−0.019	0.013
Surv1	−0.031	0.048	−0.266 ***	0.009	−0.028 *	0.013
Surv3	−0.128 *	0.060	−0.457 ***	0.042	−0.128 *	0.060
Surv5	−0.101 *	0.051	−0.377 ***	0.035	−0.101 *	0.051
Surv7	−0.058	0.033	−0.154 ***	0.023	−0.058	0.033
Surv9	−0.029	0.017	−0.052 ***	0.012	−0.029	0.017
Cul_CL_	−0.015	0.044	−0.097 **	0.030	−0.015	0.044
Cul_INF_	−0.012	0.057	0.399 ***	0.040	0.012	0.057
Cul_UD_	0.028	0.043	−0.125 ***	0.030	0.028	0.043

*** *p* < 0.001, ** *p* < 0.01, * *p* < 0.05. ^1^ calculated with model 1, ^2^ calculated with model 3

## Data Availability

Restrictions apply to the availability of these data. Data were obtained from the Landeskuratorium der Erzeugerringe für tierische Veredelung in Bayern e.V. and are available from the authors with the permission of the Landeskuratorium der Erzeugerringe für tierische Veredelung in Bayern e.V.

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
