# Peer review of "Impact of Inbreeding and Ancestral Inbreeding on Longevity Traits in German Brown Cows"

_animals, 2023, doi:10.3390/ani13172765_

Round 1

Reviewer 1 Report

The paper is very well developed and the results are well described. Below I ask some questions just to learn more about the study, not to change the scope of the research.
Frankly, I do not like this type of analysis based on hundreds of thousands of records. With such a large number, everything will be statistically significant, but this large data set was collected from many animals, in many places, and from many people with different conditions, so the random effect is large and cannot be estimated. The inbreeding depression results are significant, but perhaps one could plot how these estimates were fitted to the data, R2 values are missing.
Line 187 and following: There is information on the estimation of heritability, but information on how these parameters were calculated in the methods is missing. What model was used for the calculation and how was inbreeding included or not in this parameter estimation.
For binomial traits, such as survival, one could report the average degree of inbreeding for both traits in the appendix, but from a biological point of view there is no difference, but the amount of records makes it significant.
There is no interaction between the analyzed factors
Why was logistic regression not used for binomial traits?
How was the distribution of inbreeding or other traits checked and what was used if the distribution was not normal.
Line 302 - if there is inbreeding depression on survival, it is typical for older animals to have a lower inbreeding score.
The heterosis analyzed here was only the value of the contribution of two breeds in the parents - what about the relationship between the ancestors within the same breed - this could be a reason for inbreeding, so the two values inbreeding and heterosis are independent or not
Purging is very difficult to observe and detect in a population where new genotypes are included by migration, so in this case it is not a typical closed population.

Author Response

Manuscript ID: animals-2514172

Title:  Impact of inbreeding and ancestral inbreeding on longevity traits in German Brown cows

The authors thank the reviewer and editor for their time and work to review our manuscript.

We acknowledge the very valuable comments and recommendations for improving the manuscript.

Reviewer 1

Yes

Can be improved

Must be improved

Not applicable

Does the introduction provide sufficient background and include all relevant references?

(x)

( )

( )

( )

Are all the cited references relevant to the research?

(x)

( )

( )

( )

Is the research design appropriate?

(x)

( )

( )

( )

Are the methods adequately described?

( )

(x)

( )

( )

Are the results clearly presented?

( )

(x)

( )

( )

Are the conclusions supported by the results?

(x)

( )

( )

( )

Comments and Suggestions for Authors

The paper is very well developed and the results are well described. Below I ask some questions just to learn more about the study, not to change the scope of the research.
Frankly, I do not like this type of analysis based on hundreds of thousands of records. With such a large number, everything will be statistically significant, but this large data set was collected from many animals, in many places, and from many people with different conditions, so the random effect is large and cannot be estimated. The inbreeding depression results are significant, but perhaps one could plot how these estimates were fitted to the data, R2 values are missing.

Comment: Plotting about 480,000 records against a regression line may result in dense cloud of points.

Variance explained results from b’X’y and contains less information than SE of b values and p-values. We could not find any previous report giving R2-values. The R2-values for inbreeding depression effects may be rather small due to the large effects caused by herd effects.

Line 187 and following: There is information on the estimation of heritability, but information on how these parameters were calculated in the methods is missing. What model was used for the calculation and how was inbreeding included or not in this parameter estimation

Amended:

Line 177-181: The model employed for the estimation of heritabilities included all fixed and random effects models as the above mentioned models, and the proportion of US-Brown Swiss in classes of 10% (<31%, 31-40%, 41-50%, 51-60%, 61-70%, 71-80% > 80%) [21]. Animal models with a binomial distribution function for 0/1-traits yielded very similar estimates for heritabilities and thus, we employed linear animal models.

For binomial traits, such as survival, one could report the average degree of inbreeding for both traits in the appendix, but from a biological point of view there is no difference, but the amount of records makes it significant.
There is no interaction between the analyzed factors.

Amended:

We have added Table S5 showing the differences due to inbreeding depression for traits analysed using the b-values estimated.

Why was logistic regression not used for binomial traits?

Comment:

In routine estimation of breeding values binomial distribution are not used and therefore we wanted reporting the results of the linear scale. Most reports say that differences are rather small and may have no effects. We estimated heritabilities using a binomial distribution function in subsamples of the data and achieved very similar heritability estimates. Thus, we did not follow up models with a binomial distribution function. Previous reports on inbreeding depression also used linear models. Comparing data with these reports is easier when using linear models.

How was the distribution of inbreeding or other traits checked and what was used if the distribution was not normal.

Comment:

In this study, the general level of inbreeding was low, but nevertheless there are always highly inbred animals in a population leading to a skewed distribution. High maximum levels of inbreeding have also been observed e.g. in Martikainen, 2017 (Fmax = 0.29), Tohidi, 2023 (Fmax = 0.38), Hinrichs, 2015 (Fmax = 0.39), Doekes, 2019 (Fmax = >15) Thus, inbreeding coefficients are not expected to be normally distributed. To our knowledge, there is no literature that accounted for unequal distribution of inbreeding, when inbreeding depression was discussed. In Table S7 we provide inbreeding coefficients from previous studies in Holsteins.

Amended:

We included Supplementary Table S1 displaying the descriptive statistics of the inbreeding coefficients.

Line 302 - if there is inbreeding depression on survival, it is typical for older animals to have a lower inbreeding score.

Amended:

Line 341-343:

Thus, cows surviving to higher lactation numbers are expected to show decreasing inbreeding coefficients (Supplementary Table S5c, Supplementary Table S6).

The heterosis analyzed here was only the value of the contribution of two breeds in the parents - what about the relationship between the ancestors within the same breed - this could be a reason for inbreeding, so the two values inbreeding and heterosis are independent or not?

Amended:

We added Table S2 showing correlation coefficients among all different coefficients. We can conclude that heterosis and inbreeding coefficients are close to zero and not positively correlated and thus not influenced by collinearity.

In addition, we calculated models with just one inbreeding effect to prove consistency of the model outcomes. In our data, these outcomes were very consistent.

See lines 420-424

Considering the definition of Ballou (1997) [12], a significant favorable regression coefficient of the interaction of F with Fa_Bal would provide evidence for purging, while this study revealed significant negative effects for HL, LPL, NC and lifetime production. Moreover, the effects of Fa_Bal were also negative when considered in the model as a regression coefficient alone.

Lines 436-440:

In addition, inclusion of Ahc alone or simultaneously with F resulted in similar estimates for the regression coefficients of Ahc. We also tested the interaction of Ahc and F using model 2, which revealed comparable results to the interaction of Fa_Bal and F. In general, results for Fa_Bal and Ahc were very similar in this study with slightly higher estimates of Fa_Bal for some of the traits when included in the different models,   

Purging is very difficult to observe and detect in a population where new genotypes are included by migration, so in this case it is not a typical closed population.

Comment:

We agree with the reviewer. However, models applied account for heterosis and recombination effects. In addition, the new male genotypes were crossbred and inbreeding was raising with increasing US-Brown Swiss proportions. The population under analysis does not consist of animals from an F1-generation.

Reviewer 2 Report

The reviewed study addresses the possibility of purging affecting phenotypes in German Brown cows. The article is reasonably well written and straightforward to understand. This topic, in one way or another, has been of occasional interest for a long time (e.g., Dickerson, 1973. Proc. Anim. Brdg. Genet. Symp. in honor of J.L. Lush). In a selection experiment with quail, MacNeil and others (DOI: 10.1007/BF00263401; doi: 10.1007/BF00263402) provide evidence for purging to increase population fitness both through a reduction in the magnitude of inbreeding depression over time and when the populations under study were subjected to continuous inbreeding. Preferential referencing of relatively recent literature ignores this historical background.

There are three primary issues in the analysis that might affect the ability to detect the effects of interest. The opportunity to detect the inbreeding effects may be limited due to a lack of variation. 1) To what extent are the effects of interest collinear with other independent variables in the linear models? 2) How much variance exists in the ancestral levels of inbreeding? 3) How much variance is there in recent inbreeding after accounting for the historical inbreeding?

The data set used for this study is rather dated (ending in 2001). More recent records may provide greater opportunity to detect the effects of interest. In addition, since longevity is a primary phenotype of interest, it might be appropriate to use survival analysis methodology that simultaneously accommodates censored records. 

Author Response

Manuscript ID: animals-2514172

Title:  Impact of inbreeding and ancestral inbreeding on longevity traits in German Brown cows

The authors thank the reviewer and editor for their time and work to review our manuscript.

We acknowledge the very valuable comments and recommendations for improving the manuscript.

Reviewer 2

Yes

Can be improved

Must be improved

Not applicable

Does the introduction provide sufficient background and include all relevant references?

( )

(x)

( )

( )

Are all the cited references relevant to the research?

( )

(x)

( )

( )

Is the research design appropriate?

( )

( )

(x)

( )

Are the methods adequately described?

(x)

( )

( )

( )

Are the results clearly presented?

(x)

( )

( )

( )

Are the conclusions supported by the results?

( )

(x)

( )

( )

The reviewed study addresses the possibility of purging affecting phenotypes in German Brown cows. The article is reasonably well written and straightforward to understand. This topic, in one way or another, has been of occasional interest for a long time (e.g., Dickerson, 1973. Proc. Anim. Brdg. Genet. Symp. in honor of J.L. Lush). In a selection experiment with quail, MacNeil and others (DOI: 10.1007/BF00263401; doi: 10.1007/BF00263402) provide evidence for purging to increase population fitness both through a reduction in the magnitude of inbreeding depression over time and when the populations under study were subjected to continuous inbreeding. Preferential referencing of relatively recent literature ignores this historical background.

Amended:

Thank you very much for recommending this historical literature. We have referred to it in the introduction.

  1. 74-80: “According to Dickerson [9], the response to selection is stronger in mating systems with ful sib mating compared to random mating systems. This has already been shown in former breeding experiments on quails, where the populations with inbreeding in former generations were less susceptible to inbreeding depression then those, having no history of inbreeding. Additionally, the reproductive performance after some generations of intensively inbred populations succeeded those of the random mating population [10,11].”

There are three primary issues in the analysis that might affect the ability to detect the effects of interest. The opportunity to detect the inbreeding effects may be limited due to a lack of variation.

To what extent are the effects of interest collinear with other independent variables in the linear models?

Amended:

We added Table S2 showing correlation coefficients among all different coefficients. We can conclude that heterosis and inbreeding coefficients are close to zero and not positively correlated and thus not influenced by collinearity.

How much variance exists in the ancestral levels of inbreeding?

Amended:

We included Supplementary Tables S2 and S5, to show means, standard deviations, minimum and maximum of the inbreeding coefficients of data analysed. The variance of ancestral inbreeding coefficients (Relation of SD to Mean) is comparable to the study on Iranian Holstein (Tohidi, 2023). Table S7 contains SD of inbreeding coefficients from previous studies and our data contain variation in inbreeding coefficients comparable to these previous studies. Even in a study with high inbreeding coefficients (Doekes et al. 2019), SD was not larger than in our study.

To illustrate the effects that come along with changing ancestral inbreeding, we additionally calculated the differences of expected phenotypes between low and high ancestral inbreeding levels expressed as the 5th and 95th percentile, corresponding to 0.02 years and 0.03 (7 and 11 days) and 125 kg and 174 kg LMY for Fa_Kal and Ahc (Supplementary Table S5).

Amended:

Line 271-275:

The expected phenotypes of high (95% percentile) and low (5% percentile) inbred cows were calculated to further illustrate the effect of changing inbreeding coefficients. For classical inbreeding, the differences corresponded to 33 days for HL and 873 kg LMY. Differences in ancestral inbreeding levels were lower with 7 and 11 days of HL and 125 and 174 kg LMY for Fa_Kal and Ahc, respectively (Supplementary Table S5).

How much variance is there in recent inbreeding after accounting for the historical inbreeding?

Comment:

Recent and ancestral inbreeding were estimated simultaneously in Model 4. In this case FNew was close to zero and p-values did not reach significance levels. In our models, the significance of the effects is characterized by the P-values and the estimated standard errors of the regression coefficients. Calculating F- and t-statistics provides more information than variance explained, particularly in models with a large proportion of variance caused by other effects. We could not find any paper containing the variance explained by regression coefficients for inbreeding. We did not consider random regression for inbreeding.

The data set used for this study is rather dated (ending in 2001). More recent records may provide greater opportunity to detect the effects of interest. In addition, since longevity is a primary phenotype of interest, it might be appropriate to use survival analysis methodology that simultaneously accommodates censored records. 

Survival analysis was beyond our scope in this manuscript. We thank the reviewer for this valuable advice. This may be an issue in a follow up study.

Amended:

Line 468-484:

The overall level of inbreeding and ancestral inbreeding in the current population was not high due to introgression of US-Brown Swiss bulls, thus purging might possibly occur in future generations, where a further increase in inbreeding has been observed [19]. Additional analysis of survival traits using birth year cohorts from 2002-2008 and records through 2018 for Fa_Kal and FNew gave estimates for Surv1 of 0.0998±0.3412 and -0.1168±0.0522 with p-Values of 0.7698 and 0.0251, respectively, and for Surv3 estimates of 0.0159±0.4229 and -0.11515±0.0646 with p-Values of 0.9699 and 0.07488, respectively. Mean values and standard deviations for Fa_Kal and FNew were 0.00183±0.00264 and 0.01569±0.01695, respectively. These preliminary data suggest that significant purging effects are not yet present, even though positive effects for Fa_Kal were observed. On the other hand estimates for F (Surv1: -0.2790±0.0438, p-Value <0.001 and Surv3: -0.5821±0.0548, p-Value <0.001) and FNew indicated larger negative effects due to inbreeding depression. A similar picture emerged for effective lifetime performance for milk, fat and protein yield in this dataset with positive non-significant estimates for Fa_Kal and negative non-significant estimates for FNew. Purging may even not counterbalance the negative effects of new inbreeding in these data from 2002-2008. 

Line 529-532:

As the present data have some limitations in terms of the depth of the pedigree and the endpoint of the data (2001), future studies should possibly use censored data and survival analysis methods to gain insights into the actual development of longevity traits.

Reviewer 3 Report

The manuscript is well-structured, however, I believe there are a few areas that could benefit from additional clarity or exploration:

  1. Further elaboration on the mechanisms causing negative effects of ancestral inbreeding on traits would enhance understanding. Discussing genetic contributions from different ancestral generations and their potential interaction with selection practices could provide valuable insights.
  2. Comparing the absence of purging effects and the influence of ancestral inbreeding on longevity traits with other dairy cattle breeds in similar contexts could strengthen the conclusions.
  3. Exploring the historical selection pressure on productivity versus functional traits such as fertility and longevity would offer a deeper understanding of the observed effects.
  4. Addressing breed-specific factors that might explain the absence of purging effects in German Brown cattle would enrich the discussion.
  5. Exploring the genetic factors that connect longevity and fertility traits would contribute to a more holistic understanding of inbreeding effects.
  6. Considering the long-term genetic implications of the findings for breeding strategies and genetic improvement programs could be valuable.
  7. Discussing potential limitations or assumptions in the models used for analysis would enhance the robustness of the conclusions.
  8. Proposing avenues for future research could provide a more comprehensive understanding of the interplay between inbreeding, purging, and longevity.
  9. Highlighting practical implications for breeding programs and herd management would add practical value to the study's findings.
  10. Acknowledging the limitations of projecting results to future generations and potential external factors impacting observed trends would provide a balanced perspective.

Kindly review the manuscript to ensure the coherence and completeness of the sentences. For instance, please review Lines 31 - 33:  "... the survival to 2nd, 4th, 6th, 8th, 10th ... ".

Author Response

Manuscript ID: animals-2514172

Title:  Impact of inbreeding and ancestral inbreeding on longevity traits in German Brown cows

The authors thank the reviewer and editor for their time and work to review our manuscript.

We acknowledge the very valuable comments and recommendations for improving the manuscript.

Reviewer 3

Yes

Can be improved

Must be improved

Not applicable

Does the introduction provide sufficient background and include all relevant references?

(x)

( )

( )

( )

Are all the cited references relevant to the research?

(x)

( )

( )

( )

Is the research design appropriate?

(x)

( )

( )

( )

Are the methods adequately described?

(x)

( )

( )

( )

Are the results clearly presented?

( )

(x)

( )

( )

Are the conclusions supported by the results?

( )

(x)

( )

( )

The manuscript is well-structured, however, I believe there are a few areas that could benefit from additional clarity or exploration:

Further elaboration on the mechanisms causing negative effects of ancestral inbreeding on traits would enhance understanding. Discussing genetic contributions from different ancestral generations and their potential interaction with selection practices could provide valuable insights.

Amended:

Line 299-302:

Models regarding ancestral inbreeding coefficients revealed significant negative effects, and thus we were not able to demonstrate purging effects. Under purging ancestral inbreeding would have been expected to exert positive effects on the traits.

Comparing the absence of purging effects and the influence of ancestral inbreeding on longevity traits with other dairy cattle breeds in similar contexts could strengthen the conclusions.

Amended:

We did not find any literature of other dairy breeds, dealing with ancestral inbreeding on longevity based on pedigree data. See also Table S7.

Exploring the historical selection pressure on productivity versus functional traits such as fertility and longevity would offer a deeper understanding of the observed effects.

Amended:

L497-502:

In the German Brown, the productivity traits fat and protein yield and protein percentage were economically weighted with 48% until 2015, resulting in high selection pressure for these traits. In contrast economical weights for functional traits as longevity and fertility were only 16.1% and 8.6%, indicating less intensive selection. Estimation of breeding values for longevity started in 1995, so these traits had less time to express purging.  

Addressing breed-specific factors that might explain the absence of purging effects in German Brown cattle would enrich the discussion.

Amended:

Line 328-338:

In order to evaluate contributions from US-Brown Swiss genes, we also employed a model with F and its interaction with classes of US-Brown Swiss genes. These analyses showed for all traits under analysis, that cows with less than 50% US-Brown Swiss genes had less inbreeding depression and inbreeding depression increased up to 70-80% US-Brown Swiss genes and then decreased with even higher proportions of US-Brown Swiss genes. This may indicate that negative effects of inbreeding seem related with proportions of US-Brown Swiss genes and the strategy of using bulls with different proportions of US-Brown Swiss genes. Introducing of US-Brown Swiss bulls may be associated with less inbreeding depression, because these US-Brown Swiss bulls may not be so closely related with the German population in comparison than German Brown bulls with less than 100% US-Brown Swiss genes.

Exploring the genetic factors that connect longevity and fertility traits would contribute to a more holistic understanding of inbreeding effects.

Amended:

L385-386:

Previous studies reported positive correlations between fertility and longevity traits, further explaining the link between inbreeding, longevity and fertility [39,40].

Considering the long-term genetic implications of the findings for breeding strategies and genetic improvement programs could be valuable.

Amended:

L526-529: Our results indicate that a further decline of lifetime and lifetime production can be prevented when measures can be implemented to slow down the increase of ancestral inbreeding and the further increase of new inbreeding through US-Brown Swiss genes.

Discussing potential limitations or assumptions in the models used for analysis would enhance the robustness of the conclusions.

Amended:

Line 459- 463:

However, the informative value of the different models used strongly depends on the structure of the pedigree. Deeper pedigrees are likely to reveal more inbreeding, which could also affect the results in terms of inbreeding depression and adjustment. The depth of the pedigree in this study was comparable to other studies, as we have discussed previously [19].

Proposing avenues for future research could provide a more comprehensive understanding of the interplay between inbreeding, purging, and longevity.

Amended:

L529-532:

At the present, our data have some limitations in terms of the depth of the pedigree and the endpoint of the data (2001), future studies should possibly use censored data and survival analysis methods to gain insights into the actual development of longevity traits.

Highlighting practical implications for breeding programs and herd management would add practical value to the study's findings.

Amended:

L526-529:

Our results indicate that a further decline of lifetime and lifetime production can be prevented when measures can be implemented to slow down the increase of ancestral inbreeding and the further increase of new inbreeding through US-Brown Swiss genes.

Acknowledging the limitations of projecting results to future generations and potential external factors impacting observed trends would provide a balanced perspective.

Amended:

Line 468-484:

The overall level of inbreeding and ancestral inbreeding in the current population was not high due to introgression of US-Brown Swiss bulls, thus purging might possibly occur in future generations, where a further increase in inbreeding has been observed [19]. Additional analysis of survival traits using birth year cohorts from 2002-2008 and records through 2018 for Fa_Kal and FNew gave estimates for Surv1 of 0.0998±0.3412 and -0.1168±0.0522 with p-Values of 0.7698 and 0.0251, respectively, and for Surv3 estimates of 0.0159±0.4229 and -0.11515±0.0646 with p-Values of 0.9699 and 0.07488, respectively. Mean values and standard deviations for Fa_Kal and FNew were 0.00183±0.00264 and 0.01569±0.01695, respectively. These preliminary data suggest that significant purging effects are not yet present, even though positive effects for Fa_Kal were observed. On the other hand estimates for F (Surv1: -0.2790±0.0438, p-Value <0.001 and Surv3: -0.5821±0.0548, p-Value <0.001) and FNew indicated larger negative effects due to inbreeding depression. A similar picture emerged for effective lifetime performance for milk, fat and protein yield in this dataset with positive non-significant estimates for Fa_Kal and negative non-significant estimates for FNew. Purging may even not counterbalance the negative effects of new inbreeding in these data from 2002-2008.

Comments on the Quality of English Language

Kindly review the manuscript to ensure the coherence and completeness of the sentences. For instance, please review Lines 31 - 33:  "... the survival to 2nd, 4th, 6th, 8th, 10th ... ".

Amended at different places.

Round 2

Reviewer 2 Report

The references that were provided in the previous review were only examples of literature that was overlooked. The authors are encouraged to conduct a more thorough review of the literature that pertains to their subject matter.

The obvious questions about the opportunities to extend the dataset to more recent records remain.

Author Response

Impact of inbreeding and ancestral inbreeding on longevity traits in German Brown cows

Wirth et al.

We thank the editor and reviewer for their valuable comments and recommendations to improve our manuscript. We revised our manuscript accordingly.

Open Review

Quality of English Language

( ) I am not qualified to assess the quality of English in this paper
( ) English very difficult to understand/incomprehensible
( ) Extensive editing of English language required
( ) Moderate editing of English language required
( ) Minor editing of English language required
(x) English language fine. No issues detected

Yes

Can be improved

Must be improved

Not applicable

Does the introduction provide sufficient background and include all relevant references?

( )

(x)

( )

( )

Are all the cited references relevant to the research?

(x)

( )

( )

( )

Is the research design appropriate?

( )

(x)

( )

( )

Are the methods adequately described?

(x)

( )

( )

( )

Are the results clearly presented?

(x)

( )

( )

( )

Are the conclusions supported by the results?

( )

(x)

( )

( )

Comments and Suggestions for Authors

The references that were provided in the previous review were only examples of literature that was overlooked. The authors are encouraged to conduct a more thorough review of the literature that pertains to their subject matter.

Amended:

Lines 74-91

This means that after a few generations of inbreeding, the best performing individuals survive and reach the performance level of the non-inbred or less inbred individuals, or even achieve higher performance, while the poorer performing individuals carrying deleterious alleles die or do not reproduce, so that these alleles are eliminated from the population [9,10]. According to Dickerson [11], the response to selection is stronger in mating systems with complete sibling mating than in systems with random mating. To date, the presence of purging has been studied in various experiments, including self-fertilisation in plants [12], sibling mating in animals [13-16], mainly examining traits related to life history [17], or in small captive populations [18,19]. For example, in a previous breeding experiment with quails, it was shown that populations with inbreeding in previous generations were less prone to inbreeding depression than populations without inbreeding history. Furthermore, the reproductive performance of populations with intensive inbreeding was better after a few generations than that of populations with random mating [13,20]. However, the intensity of purging depends on the extent to which a trait is affected by deleterious alleles and how these alleles affect fertility and survival. Lethal or semi-lethal alleles are more easily eliminated than less deleterious ones [8,9,17].

Lines 601-620

  1. Hedrick, P.W.; Garcia-Dorado, A. Understanding inbreeding depression, purging, and genetic rescue. Trends in ecology & evolution 2016, 31, 940-952, doi:http://dx.doi.org/10.1016/j.tree.2016.09.005.
  2. Hedrick, P.W. Purging inbreeding depression and the probability of extinction: full-sib mating. Heredity 1994, 73, 363, doi:https://doi.org/10.1038/hdy.1994.183.
  3. Howard, J.T.; Pryce, J.E.; Baes, C.; Maltecca, C. Invited review: Inbreeding in the genomics era: Inbreeding, inbreeding depression, and management of genomic variability. J Dairy Sci 2017, 100, 6009-6024, doi:10.3168/jds.2017-12787.
  4. Dickerson, G.E. Inbreeding and heterosis in animals. J Anim Sci 1973, 1973, 54-77, doi:10.1093/ansci/1973.Symposium.54.
  5. Byers, D.L.; Waller, D.M. Do Plant Populations Purge Their Genetic Load? Effects of Population Size and Mating History on Inbreeding Depression. Annual Review of Ecology and Systematics 1999, 30, 479-513, doi:10.1146/annurev.ecolsys.30.1.479.
  6. MacNeil, M.; Kress, D.; Flower, A.; Blackwell, R. Effects of mating system in Japanese quail: 2. Genetic parameters, response and correlated response to selection. Theoretical and applied genetics 1984, 67, 407-412.
  7. Pérez-Pereira, N.; Pouso, R.; Rus, A.; Vilas, A.; López-Cortegano, E.; García-Dorado, A.; Quesada, H.; Caballero, A. Long-term exhaustion of the inbreeding load in Drosophila melanogaster. Heredity 2021, 127, 373-383, doi:10.1038/s41437-021-00464-3.
  8. Frankham, R.; Gilligan, D.M.; Morris, D.; Briscoe, D.A. Inbreeding and extinction: Effects of purging. Conservation Genetics 2001, 2, 279-284, doi:10.1023/A:1012299230482.
  9. López‐Cortegano, E.; Vilas, A.; Caballero, A.; García‐Dorado, A. Estimation of genetic purging under competitive conditions. Evolution 2016, 70, 1856-1870, doi:10.1111/evo.12983.
  10. Crnokrak, P.; Barrett, S.C. Perspective: purging the genetic load: a review of the experimental evidence. Evolution 2002, 56, 2347-2358, https://doi.org/10.1111/j.0014-3820.2002.tb00160.x.

The obvious questions about the opportunities to extend the dataset to more recent records remain.

Amended:

Line 494-500

Other opportunities for including survival in more recent data would be multiple trait animal models that only regard survival up to the first four lactation numbers, but split the first or each lactation into three different periods to account for the distribution of reasons for culling in the different lactation periods. These multiple trait animal models need to be tested to see whether they can be extended to cows surviving more than seven or nine lactations if records from the younger birth cohorts are incomplete [52-54].

Line 545-549

At the present, our data have some limitations in terms of the depth of the pedigree and the endpoint of the data (2001), future studies should possibly use censored data and survival analysis methods or multiple trait animal models for survival to different parities and periods within the respective lactations to gain insights into the actual development of longevity traits.